# Relative Consolidation of the Kappa Variant Pre-Dates the Massive Second Wave of COVID-19 in India

**DOI:** 10.3390/genes12111803

**Published:** 2021-11-16

**Authors:** Jitendra Singh, Anvita Gupta Malhotra, Debasis Biswas, Prem Shankar, Leena Lokhande, Ashvini Kumar Yadav, Arun Raghuvanshi, Dipesh Kale, Shashwati Nema, Saurabh Saigal, Sarman Singh

**Affiliations:** 1Translational Medicine Centre, All India Institute of Medical Sciences, Bhopal 462020, India; jattinsingh@gmail.com; 2Department of Microbiology, All India Institute of Medical Sciences, Bhopal 462020, India; anvitagupta16@gmail.com (A.G.M.); debasis.microbiology@aiimsbhopal.edu.in (D.B.); premshankar.1506@gmail.com (P.S.); aleenashikha@gmail.com (L.L.); ashvini.microbiology@aiimsbhopal.edu.in (A.K.Y.); contact.arun.micro@gmail.com (A.R.); dipsh21187@gmail.com (D.K.); snema.microbiology@aiimsbhopal.edu.in (S.N.); 3Department of Anesthesia and Critical Care, All India Institute of Medical Sciences, Bhopal 462020, India; saurabh.criticalcare@aiimsbhopal.edu.in

**Keywords:** COVID-19, SARS-CoV-2 genome, next generation sequencing, ion torrent, alpha variant, kappa variant and delta variant

## Abstract

India experienced a tragic second wave after the end of March 2021, which was far more massive than the first wave and was driven by the emergence of the novel delta variant (B.1.617.2) of the SARS-CoV-2 virus. In this study, we explored the local and national landscape of the viral variants in the period immediately preceding the second wave to gain insight into the mechanism of emergence of the delta variant and thus improve our understanding of the causation of the second wave. We randomly selected 20 SARS-CoV-2 positive samples diagnosed in our lab between 3 February and 8 March 2021 and subjected them to whole genome sequencing. Nine of the 20 sequenced genomes were classified as kappa variant (B.1.617.1). The phylogenetic analysis of pan-India SARS-CoV-2 genome sequences also suggested the gradual replacement of the α variant with the kappa variant during this period. This relative consolidation of the kappa variant was significant, since it shared 3 of the 4 signature mutations (L452R, E484Q and P681R) observed in the spike protein of delta variant and thus was likely to be the precursor in its evolution. This study demonstrates the predominance of the kappa variant in the period immediately prior to the second wave and underscores its role as the “bridging variant” between the α and delta variants that drove the first and second waves of COVID-19 in India, respectively.

## 1. Introduction

As of 1 August 2021, the ongoing COVID-19 pandemic had infected more than 200 million individuals and resulted in more than 4 million deaths globally. This relentless progression has been marked by periodic waves of extensive transmission, catalyzed by the emergence of novel viral variants, which have been controlled through implementation of medical and social countermeasures like vaccination, lockdown, social distancing and masking. Consistent with this pattern of alternating peaks and troughs of viral spread, India experienced the peak of its first wave in the month of September 2020, and, in the next four months, the country witnessed >80% drop in cases with the daily number of new cases dropping from 83,809 to 21,822 cases between 15 September and 31 December 2020 [1]. This dip in the number of cases continued till the middle of March 2021 (26,291 new cases reported on 15 March 2021), following which the second wave of COVID-19 emerged by the end of March 2021. Between 28 March and 30 June 2021, the average daily number of new cases across the country rose to 194,105. While India ranked third in the country-wise contribution to daily cases in January 2021, this sudden surge catapulted India to the topmost position by the first week of April 2021 and heralded the onset of the colossal second wave in the country [2]. Considering the reports of novel variants of the SARS-CoV-2 virus driving new waves of the pandemic in the UK, Brazil, and South Africa, we were interested in exploring if the second wave in India was also orchestrated by the emergence of a new viral variant [3,4,5,6].

We sought to unravel the dynamics of variant distribution in the period immediately preceding the second wave. Given the vastness and heterogeneity of the country, we attempted to address this question both in the local and national context. To determine the profile of variants at the local level, we undertook whole genome sequencing of 20 randomly selected samples reported from our lab in the month prior to the onset of the second wave (between 3 February and 8 March 2021). Prior to this, there had been a relative lack of data on the mutational landscape of SARS-CoV-2 strains circulating in the central Indian state of Madhya Pradesh, where this study was conducted. Similarly, to map the variant distribution at the national level during the same time span, we analyzed all SARS-CoV-2 whole genome sequences submitted to the GISAID (Global Initiative on Sharing All Influenza Data, Available online: https://www.gisaid.org/ (accessed on 9 August 2021) database from across the country. Comparing this profile of variants, with the same encountered after the first wave and that observed during the second wave, we report here that the massive second wave in India was pre-dated by a relative consolidation of the kappa variant which overhauled the erstwhile α variant and acted as a harbinger of the delta variant; and the delta variant subsequently drove the second wave and swiftly progressed to become the dominant variant of the pandemic.

## 2. Materials and Methods

To analyze whether the dramatic rise in the number of COVID-19 cases since the end of March 2021 was preceded and orchestrated by the emergence of novel variants of SARS-CoV-2 virus, we performed Whole Genome Sequencing of randomly selected samples reported from our lab in the period just prior to the escalation of cases, i.e., between 3 February and 8 March 2021 (Figure 1). By focusing on the period of transition between the two waves, we attempted to gain insight into the causation of the massive second wave.

To understand the sequence identity of strains circulating in our region immediately prior to the second wave, we randomly selected twenty known COVID-19 positive nasopharyngeal samples from the repository of specimens received in our laboratory for RT-PCR testing between 3 February 2021 and 8 March 2021. A volume of 250–300 μL of each sample was used for viral RNA extraction using the QIAmp^®^ Viral RNA mini kit (Qiagen, Hilden, Germany), as per the manufacturer’s instructions. The extracted RNA was quantified using an Invitrogen Thermo Fischer Qubit^TM^ RNA HS Assay kit on Qubit^TM^ 4.0 fluorometer (Invitrogen, Waltham, MA, USA). The assay was performed on DNA-free viral RNA concentration ranging from 3.02–8.87 ng/μL.

Reverse transcription of the extracted RNA samples was performed using SuperScript^TM^ VILO^TM^ cDNA Synthesis Kit (Invitrogen, Waltham, MA, USA) as per the kit instructions. Following reverse transcription, the cDNA was subjected to manual library preparation workflow for a 2-pool RNA panel using the Ion Torrent™ Ion AmpliSeq^TM^ Library Kit Plus. The amplicon prepared was subjected to partial digestion for barcode adaptor ligation using Ion Torrent™ Ion Xpress^TM^ Barcode Adaptor 1-96 kit. These amplicons were purified by magnetic beads of Beckman Coulter™ Agencourt^TM^ AMPure^TM^ XP Reagent. Purified amplicons were quantified with Invitrogen Thermo Fischer Qubit^TM^ dsDNA HS Assay Kit. Each sample library was diluted to 100 pM and loaded to Ion Chef^TM^ instrument (ThermoFisher Scientific, Waltham, MA, USA) for automated clonal amplification by emulsion PCR, enrichment and loading onto an Ion 530 chip. This chip was then subjected to massive parallel sequencing assay by Ion AmpliSeq ^TM^ SARS-CoV-2 Research Panel which covers > 99% of the SARS-CoV-2 genome on an Ion GeneStudio^TM^ S5 Prime Series system for complete viral genome sequencing [7].

The raw data generated by the Ion GeneStudio^TM^ S5 Prime Series system were analyzed on a Torrent Suite Server using custom plug-ins created for the Ion Ampliseq SARS CoV-2 panel. NGS QC Toolkit v 2.3.3 was used to remove low-quality and short reads. These trimmed reads were mapped to the SARS-CoV-2 reference sequence (Accession: NC_045512) and consensus sequence was generated thereafter using IRMAreport v1.3.0.2. These FASTA sequences were further processed for genome annotation and strain classification. Variant Caller v5.10.1.19 and COVID19AnnotateSnpEff were used to detect variants and annotate variants, respectively.

Principal Component Analysis (PCA) was done to cluster the sequenced genomes based on their evolutionary linkage. For PCA, the multiple sequence alignment of the sequenced samples was done in Jalview [8], and the distance matrix was then exported to R [9] for further calculations. The top two informative principal components were considered to generate the 2D scatter plot of the 20 samples.

The clade and lineages of the sequenced genomes were done using GISAID [10] and Pangolin [11] (Phylogenetic Assignment of Named Global Outbreak LINeages) respectively for geographic classification of the sequenced variants. Nextstrain [12] (available online: https://clades.nextstrain.org/ (accessed on 9 August 2021) was used for quality assessment and phylogenetic placement. Phylogenetic analysis of the WGS data of samples was performed by multiple sequence alignment using the MUSCLE program available in MEGA v10.1.7, and the phylogenetic tree was reconstructed using the neighbor-joining method and Kimura 2 parameter as the nucleotide substitution model with 1000 bootstrap values. This alignment was mapped on the global all SARS-CoV-2 sequence alignment in the Nextclade database to determine the lineage and clade of the 20 sequenced samples.

## 3. Results

To understand the contextual background of the circulating SARS-CoV-2 strains in our region during the above-mentioned transition phase, we undertook an analysis of the temporal distribution of SARS-CoV-2 variants reported from India in the GISAID database between: (a) September and December 2020 (the period covering the first wave); (b) January and mid-March 2021 (the period preceding the second wave); and (c) from mid-March to June 2021 (the period coinciding with the second wave).

The overall distribution of the different variants across India during these periods revealed that the B.1.1.7 (α) variant was predominant during the first wave, while the second wave was characterized by an overwhelming excess of B.1.617.2 (delta) variant and a corresponding decrease in the α variant (drop from 80% to 10% of pan-India sequences submitted in GISAID database). The period preceding the second wave was marked by similar proportions of B.1.617.1 (kappa) and α variants (40% and 42%, respectively), indicating a gradual replacement of the latter by the former variant during this duration (Figure 2). Notably the proportion of the delta variant, which was the dominant variant during the second wave, was unchanged between September 2020 and mid-March 2021, i.e., till the emergence of the second wave.

Going by specific timepoints, significant upsurge in the circulation of the kappa variant was observed across India since early February 2021, while towards the end of February 2021 the delta variant started intensifying and the α variant stabilized (Figure 3a). Although the overall proportion of the delta variant across the country was unchanged till the advent of the second wave, marked heterogeneity was observed in its circulation across the states. Of the 260 sequences reported for the delta variant from all over the country between 1 January and 15 March 2021, 84 (32.3%) were reported from the state of Maharashtra. The daily number of new cases during this period was also recorded to be significantly higher in Maharashtra compared to the rest of the country, thereby hinting at the increased transmissibility of the delta variant (Figure 3b).

In this study, the 20 libraries were sequenced on two 530 ion chips, consisting of 9 and 11 samples, respectively. Run and Alignment summary of both the sequencing runs is shown in Table 1.

The optimal performance of the sequencing run was interpreted from an average loading of 87.85% of the addressable wells with Ion Sphere Particles (ISP) and 98% of the ISPs being represented by libraries. Considering both of the chips, the final library ISPs, excluding the polyclonal, low-quality products and adapter dimers, represented an average of 70.05% of the total library ISPs. On average, 95.65% of the reads were aligned to the SARS-CoV-2 reference sequence reflecting adequate efficiency of target amplification and sequencing specificity. The unaligned reads were represented by non-specific sequencing products or primer dimers formed during PCR amplification. The values for mean depth, vertical coverage for the aligned base pairs, ranged from 892.2 to 27,794.5 (Table 2). The sequences of 20 SARS-CoV-2 genomes were submitted to GISAID and NCBI GenBank databases; the identifiers’ details are mentioned in Table 2.

We further delineated the relatedness of the 20 selected SARS-CoV-2 sequences, reported from our lab during the transition phase between the two waves, with the reference sequences of known SARS-CoV-2 variants of concern. For an initial assessment of the overall sequence similarity of strains included in this study, we first performed Principal Component Analysis (PCA), where the first two components explain nearly 75% of the variance. Here, we observed the distribution of the strains in three distinct clusters: (a) Cluster-1 with nine samples (EPI_ISL_1972141, EPI_ISL_1972134, EPI_ISL_1972135, EPI_ISL_1972136, EPI_ISL_1972137, EPI_ISL_1972138, EPI_ISL_1972139, EPI_ISL_1972140 and EPI_ISL_1972133), (b) Cluster-2 with three samples (EPI_ISL_1972130, EPI_ISL_1972132 and EPI_ISL_1972131) and (c) Cluster-3 with six samples (EPI_ISL_1972142, EPI_ISL_1972143, EPI_ISL_1972144, EPI_ISL_1972145, EPI_ISL_1972146 and EPI_ISL_1972147). Two of the 20 samples did not cluster with other samples and occurred as outliers in the PCA (EPI_ISL_3305853, EPI_ISL_3316398) (Figure 4).

The phylogenetic analysis of the whole genomes (from GISAID database) showed that these 20 genomes sequenced in the current study were distributed in three clades. Nine of the 20 sequences located in cluster 1 in PCA (EPI_ISL_1972141, EPI_ISL_1972134, EPI_ISL_1972135, EPI_ISL_1972136, EPI_ISL_1972137, EPI_ISL_1972138, EPI_ISL_1972139, EPI_ISL_1972140, EPI_ISL_1972133), aligned with the reference genome of the kappa variant. The three samples in cluster 2 (EPI_ISL_1972130, EPI_ISL_1972132 and EPI_ISL_1972131) aligned with the α variant, while seven sequenced samples belonged to B.1.36 lineage. Only one sample (EPI_ISL_3316398) was found close to the theta (21E) reference genome and belong to the lineage B.1.306. (Figure 5) Thus, we observe a ratio of 3:1 among the kappa and α variants in our region during the transition period between the two waves.

Having observed the predominance of the kappa variant in the period immediately preceding the massive second wave, we were interested in comparing its genomic characteristics with that of the α variant which drove the first wave of the pandemic in India. Compared to the Wuhan-Hu-1 reference sequence (GenBank accession No. NC_045512.2), a total of 39 and 24 mutations respectively were observed in the kappa and α variants sequenced in this study. The majority of these mutations were observed in ORF1ab, Spike, ORF3a and N genes comprising 88.52% of all mutations (Figure 6a). For each of these four genes, the kappa variant outnumbered the α variant in the number of mutations (Figure 6b). Except for the D614G mutation in the Spike protein, which is the defining feature of the B.1 lineage, no mutation was found to be shared between kappa and α variants. However, position 681 in the spike protein and position 203 in the N protein constituted shared loci of mutation between the two variants, though the amino acid substitutions were different. The spike protein of kappa and α variants demonstrated a total of 11 and 7 mutations respectively, of which 3 and 1 were observed in the receptor binding domain. In addition, P681R mutation in the spike protein of the kappa variant was significant in view of its location being adjacent to the furin cleavage site and thereby having the potential to influence viral entry into the host cell. While this assortment of multiple mutations in the spike protein appeared concerning, the kappa variant was noteworthy in demonstrating a relatively uniform distribution of mutations in the rest of the genome as well (Figure 6c).

## 4. Discussion

In this paper, we offer a snapshot of the relative proportion of the different SARS-CoV-2 variants circulating in India during the period prior to the emergence of the second wave of COVID-19 and report a relative consolidation of the kappa variant vis-à-vis the α variant during this transition phase. By focusing on the critical period between the two successive waves of COVID-19, our study aids in understanding of the genesis of the second wave and hints at the critical role played by the kappa variant in orchestrating the emergence of the delta variant. By virtue of sharing three critical mutations in the spike protein with the delta variant (L452R, E484Q and P681R), the kappa variant can be considered to be the “bridging variant” that acted as the forerunner of the highly transmissible delta variant that was responsible for the enormity of the second wave in India. This trend of variant emergence, reported in this first paper on the profile of SARS-CoV-2 variants from central India, is also representative of the rest of the country, as depicted in the pan-India sequences reported in the GISAID database.

Mutation analysis of the kappa variant indicates several key substitutions relative to the Wuhan-Hu-1 reference sequence (GenBank accession No. NC_045512.2). Of the 14 non-synonymous mutations observed across the genome of all the strains belonging to the kappa variant, five are found in the spike protein (E484Q, L452R, P681R, D614G, Q1071H) and among them the first two lie in the receptor binding domain (RBD) and the third one is adjacent to the furin cleavage site (Figure 7). Interestingly, mutation in position 484 is also present in the β (B.1.351) and γ (P.1) variants of concern (VOC) and eta (B.1.525) and iota (B.1.526) variants of interest (VOI). While the other variants harbor E484K as the substitution, E484Q is observed in the kappa variant [13,14] and the E484K mutation is associated with reduced susceptibility to the monoclonal antibody bamlanivimab and the combination of bamlanivimab and etesevimab [15]. Likewise, mutation in position 681 is shared with α (B.1.1.7) VOC (P681H) [16]. Similarly, L452R mutation is reported in epsilon (B.1.427 and B.1.429), which is a variant of concern reported from California [17]. This mutation is reported to increase the rate of membrane fusion and thus lead to enhanced transmissibility [18]. The kappa variant thus epitomizes a conglomeration of key sequence variations that have previously been associated with important biological properties like enhanced viral attachment, cellular fusion and reduced neutralization with serum from convalescent individuals, vaccine recipients and monoclonal antibodies [19,20,21]. Among the non-spike proteins, the kappa variant harbors the R203M mutation at the edge of the Serine/Arginine (SR) rich domain of N protein and shares sequence variations in this mutational hotspot with the α (R203M) and β (T205I) variants. This domain, considered important for viral assembly, could favor viral replication in the concerned variants [22,23]. The kappa variant thus retains several critical mutations that have been previously associated with enhanced “viral fitness,” which explains its enhanced replication efficiency and ability to overhaul the circulating α variant. Furthermore, while retaining all the key mutations contained in the kappa variant, the spike protein of the delta variant contains only one additional mutation in the RBD domain (T478K). It, thus, appears plausible that the delta variant has emerged from the kappa variant and, by dint of the additional mutation, has acquired enhanced transmissibility to drive the second wave in India and become the predominant SARS-CoV-2 variant globally.

Several recent structural studies revealed that the variants with L452R and E484Q mutations in the RBD region divulge low binding energy and achieve enhanced stability of interaction with ACE2 receptor compared to the wild type Wuhan strain [24]. Furthermore, L452R mutation is reported to reduce neutralization activity of RBD-specific monoclonal antibodies by 2–3.5 fold [25]. However, further studies need to be conducted to gain insight into the impact of the entire set of mutations present in the spike protein of kappa and delta variants on the neutralization efficiency of antibodies generated in response to the existing COVID-19 vaccines. Docking studies are also necessary to understand the interaction efficiency of the novel variants with the host receptor molecule and proteolytic enzymes and thus infer the mechanistic basis of enhanced viral replication and higher transmissibility observed with these variants.

In contrast to the majority of studies dealing with the role of the delta variant in driving the second wave of COVID-19, our study is one of the few studies that focuses on the period immediately preceding the second wave and aids in unraveling its causation. However, it suffers from the limitation of restricted sample size and lack of sufficient power to compare the disease outcome in individuals infected with the kappa and α variants. We understand that the complete absence of the delta variant in our region during the study period could be an artefact of the limited sample size. In view of the under-representation of central India in the global SARS-CoV-2 sequence database (only 1.83% of Indian sequences submitted in GISAID belong to central India), there is a need to expand the scope of this study for better characterization of the molecular epidemiology and clinical and immunological correlates of SARS-CoV-2 strains circulating in this region.

## 5. Conclusions

To summarize, this study is the first to report on the significant consolidation of the kappa variant in the period immediately preceding the second wave of the pandemic in India. We also observe the confluence of multiple mutations of biological and clinical relevance within this variant. The temporal kinetics observed in this study also suggest the potential role of the kappa variant in heralding the emergence of the delta variant that drove the second COVID-19 wave in India and swiftly emerged to become the dominant variant across the globe. As India continues to undertake the largest vaccination drive in the world, it would be interesting to monitor the evolution of SARS-CoV-2 variants in the face of mounting immune pressure emerging within the huge and heterogeneous population of the country.

## Figures and Tables

**Figure 1 genes-12-01803-f001:**
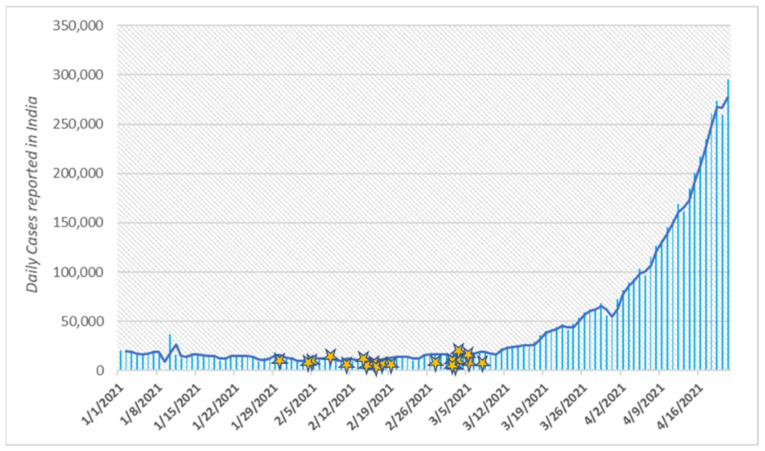
Daily count of new COVID-19 cases reported from India between January 2021 through 16 April 2021, the stars represent the collection dates of the samples included in this study.

**Figure 2 genes-12-01803-f002:**
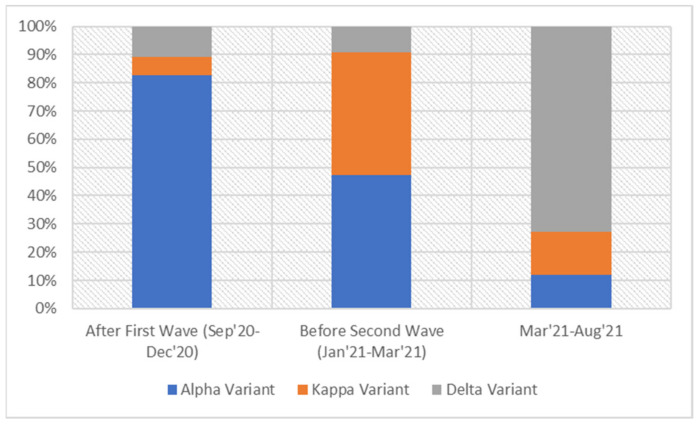
The prevalence of the α, Kappa and Delta variant illustrated in terms of the sequences of samples collected from September 2020 till the date in GISAID.

**Figure 3 genes-12-01803-f003:**
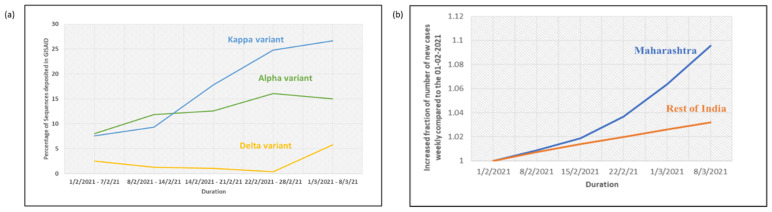
(**a**) The percentage of sequences of α, Kappa and Delta variant submitted in GISAID for clinical samples collected from February 2021 to March 2021 from India; (**b**) average increase of new cases in Maharashtra and rest of the India during the same period, relative to figures of 1 February 2021.

**Figure 4 genes-12-01803-f004:**
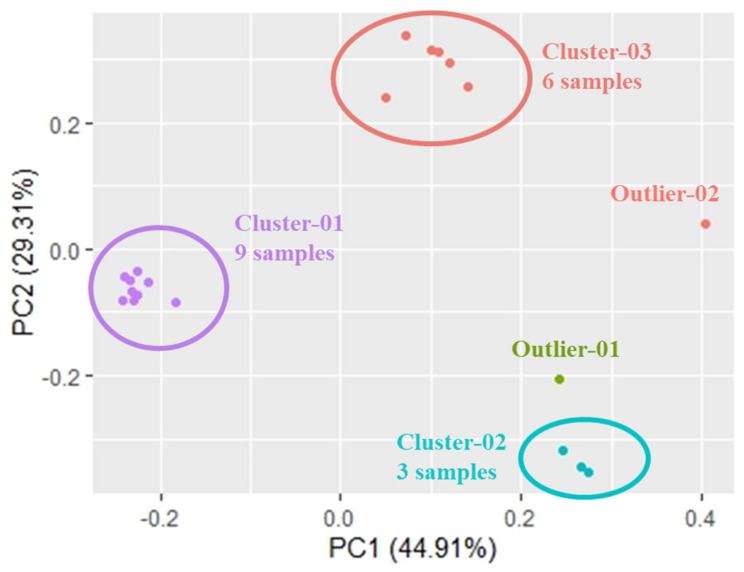
Principal Component Analysis of the 20 sequenced samples.

**Figure 5 genes-12-01803-f005:**
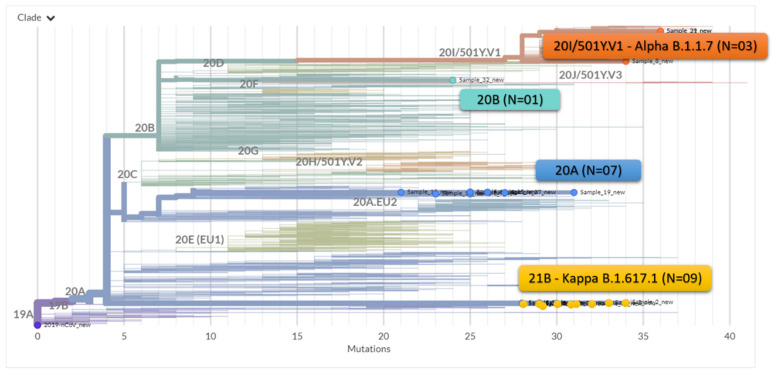
Maximum Likelihood phylogenetic tree showing the relationship of the 20 samples of SARS-CoV-2 strains in this study in the context of globally circulating strains of SARS-CoV-2.

**Figure 6 genes-12-01803-f006:**
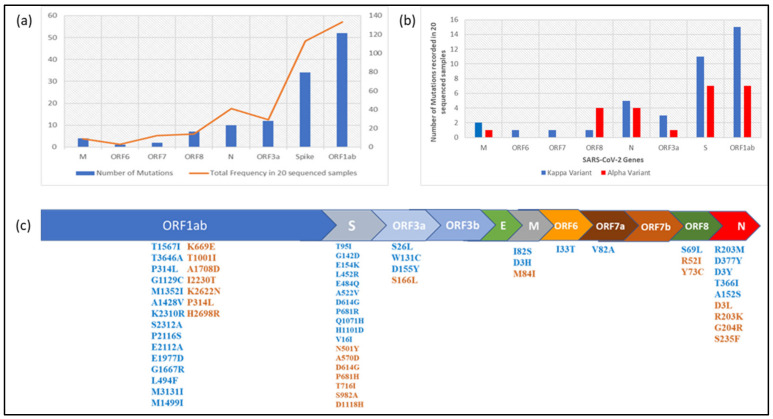
(**a**) Gene-wise depiction of number of mutations and mutation frequency in 20 sequenced samples; (**b**) clade wise mutation distribution in sequenced samples; (**c**) amino-acid mutations observed in SARS-CoV-2 genome in sequenced samples.

**Figure 7 genes-12-01803-f007:**
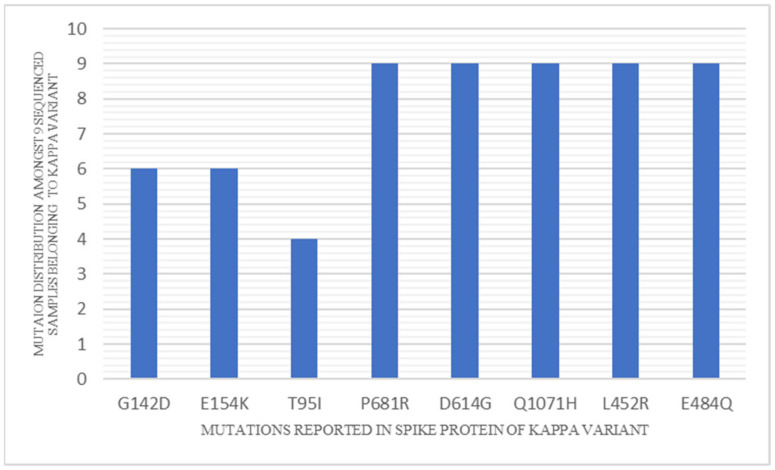
Mutation distribution among the sequenced samples belonging to the kappa variant.

**Table 1 genes-12-01803-t001:** Run and alignment summary of the two runs performed on Ion GeneStudioTM S5 System (ThermoFisher Scientific, Waltham, MA, USA) on 530 chip.

Addressable Wells	Chip-1 (9 Samples)			Chip-2 (11 Samples)		
	37,849,615			37,849,615		
With ISPs	33,700,206	89.00%		32,798,330	86.70%	
Live	33,698,044	100.00%		32,749,088	99.80%	
Test Fragments	680,001	2.00%		657,319	2.00%	
Library	33,018,043	98.00%		32,091,769	98.00%	
Library ISPs	33,018,043			32,091,769		
Filtered: Polyclonal	8,159,862	24.70%		7,284,699	22.70%	
Filtered: Low Quality	22,21,791	6.70%		1,630,141	5.10%	
Filtered: Adapter Dimer	56,745	0.20%		158,413	0.50%	
Final Library ISPs	22,579,645	68.40%		23,018,516	71.70%	
Total Reads	22,391,814			22,825,334		
Aligned Reads	21,175,935	94.60%		22,067,308	96.70%	
Unaligned Reads	1,215,879	5.40%		758,026	3.30%	
Alignment Quality	AQ17	AQ20	Perfect	AQ17	AQ20	Perfect
Total Number of Bases [Mbp]	3.83 G	3.68 G	2.96 G	4.18 G	4.02 G	3.25 G
Mean Length [bp]	193	189	158	197	192	160
Longest Alignment [bp]	377	376	359	343	343	332
Mean Coverage Depth	57.1	54.8	44.1	62.3	59.8	48.4

**Table 2 genes-12-01803-t002:** Genomic features of twenty SARS-CoV-2 clinical samples.

S.No.	Patient ID	Qbit 4.O Reading ng/ul-RNA Sample	GenBank Accession No.	GISAID Accession No.	No. of Mapped Reads	Total Assigned Amplicon Reads	% on Target Reads	Average Read per Amplicon	Mean Depth	%Uniformity	Assembly Length (bp)	PANGO Lineage	Emerging Clade	GISAID Clade
1	NCOV/AB/21/FL407	3.02	MZ562746	EPI_ISL_1972141	1,378,010	1,364,626	99.03	5758	7503	74.30%	29,837	B.1.617.1	21B (Kappa)	G
2	NCOV/AB/21/FL409	8.87	MZ562747	EPI_ISL_1972134	3,733,707	3,732,055	99.96	15,747	25,230	97.89%	29,833	B.1.617.1	21B (Kappa)	G
3	NCOV/AB/21/FL901	5.92	MZ562748	EPI_ISL_1972135	2,389,121	2,388,613	99.98	10,079	15,986	97.03%	29,834	B.1.617.1	21B (Kappa)	G
4	NCOV/AB/21/FL902	5.53	MZ562749	EPI_ISL_1972136	2,725,685	2,694,642	98.86	11,370	18,478	99.29%	29,840	B.1.617.1	21B (Kappa)	G
5	NCOV/AB/21/FL832	6.73	MZ562750	EPI_ISL_1972130	661,367	144,412	21.84	609.3	936.4	99.28%	29,816	B.1.1.7	20I/501Y.V1	GRY
6	NCOV/AB/21/FL405	3.72	MZ562751	EPI_ISL_1972142.	3,581,768	3,544,684	98.96	14,956	22,246	97.10%	29,826	B.1.36	20A	GH
7	NCOV/AB/21/EX986	3.24	MZ723921	EPI_ISL_3305853	394,294	371,849	94.31	1569	2,473	97.28%	29,800	B.1.36	20A	GH
8	NCOV/AB/20/CA052	5.65	MZ562752	EPI_ISL_1972143	4,349,146	4,289,977	98.64	18,101	28,961	99.54%	29,834	B.1.36	20A	GH
9	NCOV/AB/21/FA012	6.69	MZ562753	EPI_ISL_1972144	1,506,394	1,446,477	96.02	6103	9750	99.49%	29,848	B.1.36	20A	GH
10	NCOV/AB/21/FB757	3.73	MZ562754	EPI_ISL_1972145	4,124,452	4,104,495	99.52	17,319	26,603	96.98%	29,834	B.1.36	20A	GH
11	NCOV/AB/21/FD308	3.88	MZ562755	EPI_ISL_1972146	80,774	79,290	98.16	334.6	534.7	99.12%	29,834	B.1.36	20A	GH
12	NCOV/AB/21/FF663	3.38	MZ562756	EPI_ISL_1972137	393,314	275,581	70.07	1163	1,842	98.12%	29,833	B.1.617.1	21B (Kappa)	G
13	NCOV/AB/21/FF945	4.34	MZ562757	EPI_ISL_1972138	2,080,768	2,058,700	98.94	8686	13,990	98.64%	29,836	B.1.617.1	21B (Kappa)	G
14	NCOV/AB/21/EQ324	5.32	MZ562758	EPI_ISL_1972139	917,154	899,993	98.13	3797	5,763	97.97%	29,832	B.1.617.1	21B (Kappa)	G
15	NCOV/AB/21/FG788	3.34	MZ562759	EPI_ISL_1972140	3,939,432	3,917,148	99.43	16,528	26,628	98.21%	29,835	B.1.617.1	21B (Kappa)	G
16	NCOV/AB/21/FK354	7.18	MZ562760	EPI_ISL_1972147	1,945,720	1,901,249	97.71	8022	12,758	99.11%	29,853	B.1.36	20A	GH
17	NCOV/AB/21/FL862	4.35	MZ723922	EPI_ISL_1972132	3,313,340	3,203,191	96.68	13,516	21,669	97.89%	29,823	B.1.1.7	20I/501Y.V1	GRY
18	NCOV/AB/21/FK940	4.94	MZ562761	EPI_ISL_1972133	2,262,299	2,261,261	99.95	9541	14,961	99.10%	29,840	B.1.617.1	21B (Kappa)	G
19	NCOV/AB/21/FL880	5.27	MZ562762	EPI_ISL_1972131	2,668,661	2,667,095	99.94	11,254	18,319	99.06%	29,817	B.1.1.7	20I/501Y.V1	GRY
20	NCOV/AB/21/FL414	6.54	MZ734487	EPI_ISL_3316398	238,608	125,526	52.61	529.6	848	92.73%	29,834	B.1.306	20B	GR

## Data Availability

MDPI Research Data Policies. The genomic sequences generated in this study have been submitted in the GISAID and NCBI-GenBank database, with the accession nos. mentioned in Table 2.

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
