# Peer review of "Relative Consolidation of the Kappa Variant Pre-Dates the Massive Second Wave of COVID-19 in India"

_genes, 2021, doi:10.3390/genes12111803_

Round 1

Reviewer 1 Report

This is an interesting study about the epidemiological status of SARS-CoV-2 variants in India in the period prior to the second wave. The study is well designed, even if the sample size is small and no firm conclusions can be drawn. However, it deserves publication. Some comments to be addressed by the authors:

  1. In the section materials and methods please insert the brand of each kit used
  2. Why the authors selected only 20 samples? Were they the only swabs received in your laboratories, or was it a choice of the authors?
  3. The samples used by the authors should be independent samples; they should not be, for example, family cases because this could alter the statistics presented. Please include this information.
  4. The data shown in figure 3b could be attributable to a higher number of sequences inserted in the GSAID portal and a higher number of swabs performed from the state of Maharashtra? The authors could better explain these data
  5. Section discussion, sentence “This trend of variant emergence, reported in this first paper on the profile of SARS-CoV-2 variants from.” , please correct the punctuation.
  6. The data reported do not confirm, with certainty, that the kappa variant can be considered the precursor of the delta variant. In the discussion the authors should consider it a hypothesis.

Reviewer 2 Report

The Authors explored the local and national landscape of the viral variants in the period immediately preceding the second wave to gain insight into the mechanism of emergence of the delta variant and thus improve the causation of the second wave.

They randomly selected 20 SARS-CoV-2 positive samples and subjected them to whole genome sequencing - 9 were classified as kappa variant (B.1.617.1). The phylogenetic analysis suggested the gradual replacement of the alpha variant with the kappa variant during analyzed period.

Authors emphasize that 3 signature mutations (L452R, E484Q and P681R) in the spike protein of delta variant were probably the precursor in evolution of SARS CoV-2.

The authors write that this is the first analysis of this type in India.

They emphasize the importance of multiple mutations in the evolution of the delta variant, which then spread throughout the world. Currently in Poland there is a sharp increase in infections (fourth wave) caused in 95% by the delta variant.

In my opinion, this paper is interesting and  shows how multiple mutations affect the emergence of new variants of SARS CoV-2.

Author Response

We thank the reviewer for the appreciative comments.